# NT-proBNP Levels Influence the Prognostic Value of Mineral Metabolism Biomarkers in Coronary Artery Disease

**DOI:** 10.3390/jcm11144153

**Published:** 2022-07-17

**Authors:** Juan Martínez-Milla, Álvaro Aceña, Ana Pello, Marta López-Castillo, Hans Paul Gaebelt, Óscar González-Lorenzo, Nieves Tarín, Carmen Cristóbal, Luis M. Blanco-Colio, José Luis Martín-Ventura, Ana Huelmos, Andrea Kallmeyer, Joaquín Alonso, Carlos Gutiérrez-Landaluce, Lorenzo López Bescós, Jesús Egido, Ignacio Mahíllo-Fernández, Óscar Lorenzo, María Luisa González-Casaus, José Tuñón

**Affiliations:** 1Department of Cardiology, IIS-Fundación Jiménez Díaz, Avda. Reyes Católicos 2, 28040 Madrid, Spain; j.martinez.milla@gmail.com (J.M.-M.); aacena@fjd.es (Á.A.); ampello@fjd.es (A.P.); marta.lcastillo@fjd.es (M.L.-C.); hpgaebelt@fjd.es (H.P.G.); ogonzalez@quironsalud.es (Ó.G.-L.); andrea.kallmeyer@quironsalud.es (A.K.); 2Centro Nacional de Investigaciones Cardiovasculares (CNIC), 28029 Madrid, Spain; 3CIBERCV, 28029 Madrid, Spain; lblanco@fjd.es (L.M.B.-C.); jlmartin@fjd.es (J.L.M.-V.); jegido@fjd.es (J.E.); 4Faculty of Medicine, Universidad Autónoma de Madrid, 28049 Madrid, Spain; olorenzo@fjd.es; 5Department of Cardiology, Hospital Universitario de Móstoles, 28040 Madrid, Spain; nieves.tarin@salud.madrid.org; 6Department of Cardiology, Hospital Universitario de Fuenlabrada, 28942 Madrid, Spain; carmen.cristobal@salud.madrid.org (C.C.); cgutierrezl@salud.madrid.org (C.G.-L.); 7Faculty of Medicine, Universidad Rey Juan Carlos, Alcorcón, 28922 Madrid, Spain; joaquinjalonso@gmail.com (J.A.); llbescos@secardiologia.es (L.L.B.); 8Laboratory of Vascular Pathology, IIS-Fundación Jiménez Díaz, 28040 Madrid, Spain; 9Department of Cardiology, Hospital Universitario Fundación Alcorcón, 28040 Madrid, Spain; ahuelmos@yahoo.es; 10Department of Cardiology, Hospital de Getafe, 28040 Madrid, Spain; 11Spanish Biomedical Research Centre in Diabetes and Associated Metabolic Disorders (CIBERDEM), 28029 Madrid, Spain; 12Research Unit, IIS-Fundación Jiménez Díaz, 28040 Madrid, Spain; 13Department of Laboratory Medicine, La Paz University Hospital, 28046 Madrid, Spain; imahillo@fjd.es (I.M.-F.); mlgcasaus@gmail.com (M.L.G.-C.)

**Keywords:** heart failure, stable coronary artery disease, biomarkers

## Abstract

Background. Mineral metabolism (MM) system and N-terminal pro-brain natriuretic peptide (NT-ProBNP) have been shown to add prognostic value in patients with stable coronary artery disease (SCAD). However, the influence of NT-ProBNP on the prognostic role of MM in patients with SCAD has not been shown yet. The objective of this study is to assess the influence of NT-ProBNP on the prognostic role of MM markers in patients with SCAD. Methods: We analyzed the prognostic value of MM markers (parathormone (PTH), klotho, phosphate, calcidiol (25-hydroxyvitamin D_3_), and fibroblast growth factor-23) in 964 patients with SCAD and NT-ProBNP > 125 pg/mL vs. patient with NT-ProBNP ≤ 125 pg/mL included in five hospitals in Spain. The main outcome was the combination of death, heart failure, and ischemic events (any acute coronary syndrome, ischemic stroke, or transient ischemic attack). Results: A total of 622 patients had NT-proBNP > 125 pg/mL and 342 patients had NT-ProBNP ≤ 125 pg/mL. The median follow-up was 5.1 years. In the group of NT-proBNP > 125 pg/mL, the patients were older, and there were more females and smokers than in the group of patients with normal NT-proBNP. Additionally, the proportion of patients with hypertension, atrial fibrillation, ejection fraction < 40%, cerebrovascular attack, or prior coronary artery bypass graft was higher in the high NT-proBNP group. In the high NT-proBNP patients, the predictors of poor prognosis were PTH (HR = 1.06 (1.01–1.10), *p* < 0.001) and NT-proBNP (HR = 1.02 (1.01–1.03), *p* = 0.011), along with age (HR = 1.039 (1.02–1.06), *p* < 0.001), prior coronary artery bypass graft (HR = 1.624 (1.02–2.59), *p* = 0.041), treatment with statins (HR = 0.32 (0.19–0.53), *p* < 0.001), insulin (HR = 2.49 (1.59–4.09), *p* < 0.001), angiotensin receptor blockers (HR = 1.73 (1.16–2.56), *p* = 0.007), nitrates (HR = 1.65 (1.10–2.45), *p* = 0.014), and proton pump inhibitors (HR = 2.75 (1.74–4.36), *p* < 0.001). In the NT-proBNP ≤ 125 pg/mL subgroup, poor prognosis predictors were plasma levels of non-high-density lipoprotein (non-HDL) cholesterol (HR = 1.01 (1.00–1.02), *p* = 0.014) and calcidiol (HR = 0.96 (0.92–0.99), *p* = 0.045), as well as treatment with verapamil (HR = 11.28 (2.54–50.00), *p* = 0.001), and dihydropyridines (HR = 3.16 (1.63–6.13), *p* = 0.001). Conclusion: In patients with SCAD and NT-ProBNP > 125 pg/mL, PTH and NT-ProBNP, which are markers related to ventricular damage, are predictors of poor outcome. In the subgroup of patients with NT-ProBNP ≤ 125 pgm/L, calcidiol and non-HDL cholesterol, which are more related to vascular damage, are the independent predictors of poor outcome. Then, in patients with SCAD, baseline NT-ProBNP may influence the type of biomarker that is effective in risk prediction.

## 1. Introduction

In the evolution of evidence-based medicine, biomarkers are becoming increasingly important at the diagnostic, therapeutic and prognostic levels [1,2]. “Of these N-terminal pro-brain natriuretic peptide (NT-proBNP) is commonly used for the diagnosis of heart failure” (HF) [3], but it may also predict the development of HF and death in patients with cardiovascular disorders, including those with chronic ischemic heart disease [4,5,6,7]. Accordingly, the determination of NT-proBNP levels is currently recommended in the follow-up of ischemic heart disease [8]. Patients with elevated NT-ProBNP values are at risk of developing HF regardless of their underlying heart disease and the value of their left ventricular ejection fraction [9]. Moreover, in some cases they may predict the response to cardiovascular therapies [10]. However, to our knowledge, it has not been demonstrated if NT-proBNP plasma levels may influence the prognostic value of other biomarkers.

The components of mineral metabolism (calcidiol, fibroblast growth factor-23 (FGF-23), phosphate, parathormone (PTH), and klotho) have been shown to be related to cardiovascular damage, and to predict the incidence of cardiovascular events [11,12,13,14,15,16,17]. However, while some of them, such as low calcidiol levels, may seem more related to vascular damage, others, such as FGF2-23 and PTH, seem to have a more important association with left ventricular damage [14,18]. We have recently demonstrated that some of these biomarkers add prognostic value to NT-proBNP plasma levels in patients with stable coronary artery disease [19]. However, it has not been shown if this prognostic value is similar in patients with normal NT-proBNP levels as compared to those with increased values of this biomarker. The hypothesis put forward in the present study is that NT-proBNP levels may stratify patients with coronary artery disease into two subgroups where the biomarkers related to mineral metabolism as well as those associated with inflammation may have a different prognostic value.

## 2. Methods

### 2.1. Patients

A total of 969 patients with stable coronary artery disease, who had suffered an acute coronary syndrome 6–12 months earlier, were included in this study between 2006 and 2014. These patients were part of the BACS & BAMI (Biomarkers in Acute Coronary Syndrome & Biomarkers in Acute Myocardial Infarction) studies, carried out in five hospitals in the Community of Madrid. The inclusion and exclusion criteria have been defined previously [19].

Patients had a second visit 6 to 12 months after hospital discharge, clinical variables were collected, and plasma extraction was performed. This paper reports data on clinical and analytical findings obtained at the time of this visit, relating them to data obtained during the subsequent follow-up. Plasma collection and baseline visits were performed between January 2007 and December 2014. Final visits were performed in June 2016.

### 2.2. Study Design

Clinical variables were recorded, and venous blood samples were drawn after fasting for 12 h and collected in an EDTA tube between 6 and 12 months after hospital discharge for the acute ischemic event. Blood samples were centrifuged at 2500× *g* for 10 min and the plasma was stored at −80 °C. Patients were cared for in their hospital according to care protocols. At the end of the follow-up, medical records were reviewed and patient status was confirmed by telephone contact.

The population was divided according to whether they had NT-proBNP > 125 pg/mL and we looked for variables associated with poor prognosis in each group.

The primary endpoint was the combination of acute ischemic events (any acute coronary syndrome, stroke, and transient ischemic attack), HF admission and all-cause mortality. Non-STEMI was defined as angina at rest lasting more than 20 min in the previous 24 h, or new-onset class III–IV angina, together with transient ST-segment depression or T-wave inversion on the electrocardiogram considered diagnostic by the treating cardiologist and/or troponin elevation. A diagnosis of STEMI required a compatible picture with angina of more than 20 min duration and ST elevation in two adjacent leads of the electrocardiogram with no response to nitroglycerin and troponin elevation. Stroke was defined as the rapid onset of a neurological deficit attributable to a focal vascular cause lasting more than 24 h or with evidence of new cerebral ischemic lesions on imaging studies. A transient ischemic attack was defined as a transient stroke with signs and symptoms that resolved before 24 h without acute cerebral ischemic lesions on imaging techniques. Events were ratified by at least two study investigators, together with a neurologist for the diagnosis of cerebrovascular events. Although all events were recorded for each case, patients were excluded from the Cox regression analysis after the first event. Thus, although the total number of events is also described, patients who had more than one event were computed only once for these analyses.

### 2.3. Analytical and Biomarker Studies

Plasma determinations were performed in the Nephrology Laboratory of the Gómez-Ulla Hospital and in the Biochemistry Laboratory of the Fundación Jiménez Díaz. The investigators who performed the laboratory studies were unaware of the clinical data. Plasma calcidiol levels were quantified by chemiluminescent immunoassay (CLIA) in the LIAISON XL analyzer (LIAISON 25OH-Vitamin D total Assay, DiaSorin, Saluggia, Italy); FGF23 levels were measured by an enzyme-linked immunosorbent assay that recognizes epitopes within the carboxyl-terminal portion of FGF-23 (Human FGF23, C-Term, Immutopics Inc., San Clemente, CA, USA); klotho levels were quantified by ELISA (human soluble klotho alpha assay kit, Immuno-Biological Laboratories Co., Gunma, Japan); intact PTH was analyzed by a second-generation automated chemiluminescence method (Elecsys 2010 platform, Roche Diagnostics, Mannheim, Germany); phosphate was determined by an enzymatic method (Integra 400 analyzer, Roche Diagnostics, Mannheim, Germany); high-sensitivity troponin was assessed by direct chemiluminescence (ADVIA Centaur; Siemens, Berlin, Germany); the amino-terminal portion of pro-BNP (NT-pro-BNP) was determined by immunoassay (VITROS, Orthoclinical Diagnostics, Raritan, NJ, USA); and high-sensitivity C-reactive protein (CRP) was assessed by latex-enhanced immunoturbidimetry (ADVIA 2400 Chemistry System, Siemens, Munich, Germany). Lipid, glucose and creatinine determinations were performed by standard methods (ADVIA 2400 Chemistry System, Siemens, Munich, Germany). Plasma concentrations of MCP-1 and galectin-3 were determined using commercially available enzyme-linked immunosorbent assay kits (BMS279/2, Bender MedSystems, Burlingame, CA, USA; DCP00, R&D Systems, Minneapolis, MN, USA, respectively), following the manufacturers’ instructions. The intra- and interassay coefficients of variation were 4.6% and 5.9% for MCP-1 and 6.2% and 8.3% for galectin-3, respectively. The estimated glomerular filtration rate was calculated using the CKD-EPI equation (Chronic Kidney Disease Epidemiology Collaboration equation).

### 2.4. Statistical Analysis

Quantitative data are shown as median (interquartile range) and qualitative variables are presented as percentages. The normality of the variables was tested using the Kolmogorov–Smirnov or Shapiro–Wilk test according to the sample size of each variable. To compare the baseline values between the two groups according to whether they had a NT-proBNP > 125 pg/mL, the Pearson chi-squared test or Fisher’s exact test was used for qualitative variables. In the case of quantitative variables, a Student’s *t*-test or the Mann–Whitney test was used, depending on whether the distribution was normal or not, respectively. The population was divided according to whether they had NT-proBNP > 125 pg/mL and univariate Cox regression was performed to analyze which variables were associated with the development of the different outcomes in each group. Afterwards, a multivariate regression analysis was performed in both groups including those variables that reached a *p* < 0.20 in the univariate analyses. The analyses were performed with IBM SPSS Statistics for Windows, Version 19.0. Armonk, NY, USA: IBM Corp., and were considered significant when “*p*” was less than 0.05 (two-tailed).

### 2.5. Ethics Statement

The study protocol was carried out according to the ethical guidelines of the 1975 Declaration of Helsinki and was approved by the human research committees of the institutions participating in this study: Fundación Jiménez Díaz, Hospital Fundación Alcorcón, Hospital de Fuenlabrada, Hospital Universitario Puerta de Hierro Majadahonda and Hospital Universitario de Móstoles. All patients signed the informed consent form.

## 3. Results

### 3.1. Patients

Of the 969 patients included, 5 were lost to follow-up, leaving 964 patients for analysis. The median age was 60.0 (52.0–72.0) years, 76.2% of cases were male and the median eGFR was 80.4 (65.3–93.1) mL/min/1.73 m^2^. A total of 342 (35.4%) participants had levels of NT-ProBNP ≤ 125 pg/mL and 622 (64.6%) > 125 pg/mL. The time elapsed since the previous acute coronary syndrome was 6.5 (6.2–7.6) months.

The baseline characteristics of the population according to NT-ProBNP are detailed in Table 1. Patients with NT-proBNP > 125 pg/mL were older, had a higher percentage of women and a higher rate of comorbidities (hypertension, diabetes, peripheral arterial disease, ventricular dysfunction, heart failure and atrial fibrillation). The group of patients with NTpProBNP > 125 pg/mL had A high proportion of STEMI as the index acute coronary syndrome. Analytically, in the group of NT-proBNP ≤ 125 pg/mL, the median of NT-ProBNP was 71.5 (47.2–95.8) pg/mL and, in the other group, it was 305.0 (187.7–578.0) pg/mL. Those with NT-proBNP > 125 pg/mL had higher levels of C-reactive protein, high-sensitivity troponin I, FGF23, MCP-1, Galectin-3 and PTH, with lower levels of klotho and estimated glomerular filtration rate.

### 3.2. Clinical Events

During a median follow-up of 5.39 (2.81–6.92) years, 184 patients developed the primary objective (ischemic event, HF admission or all-cause death) with a total of 278 events. Regarding the primary end point, the total number of events was higher in patients with NT-proBNP > 125 pg/mL (139 events (76%) vs. 45 (24%)) (Appendix A). There were 116 patients who developed an ischemic event and 75 died. Of the total ischemic events, 108 were acute coronary syndromes (44 unstable angina, 48 NSTEMI, and 16 STEMI). Twenty-three patients died in addition to having an ischemic event. In relation to the combined endpoint of death or ischemic event, there were a total of 168 episodes and the number of events was also higher in the group of NT-proBNP > 125 pg/mL (124 events (74%) vs. 26%)) (Appendix A).

A total of 59 episodes of HF were recorded and 96 patients developed an episode of death or heart failure. Deaths were significantly higher in group of NT-proBNP > 125 pg/mL. Additionally, episodes of HF were significantly higher in the NT-proBNP > 125 pg/mL group (57 episodes vs. 2 episodes).

Seventy-five deaths were observed during the follow-up. The cause of death was cardiovascular in 28 patients, cancer in 15, infection in 9, renal failure in 3, advanced cognitive impairment in 3, pancreatitis in 2, gastrointestinal bleeding in 2, exacerbation of pulmonary disease in 2, and other causes in 3. Eight deaths were of unknown origin.

### 3.3. Prognostic Value of the Components of Mineral Metabolism According to NT-proBNP

In the univariate COX analysis in the group with a NT-proBNP ≤ 125 pg/mL, the variables that were predictors with a *p* < 0.2 of the development of the primary objective were Caucasian, age, high blood pressure, treatment with statins, ezetimibe, betablockers, mineralocorticoid receptor antagonists, nitrates, calcium antagonists, and analytical parameters such as glucose, total cholesterol, LDL, non-HDL, calcidiol and PTH (Table 2A).

In the multivariate analysis for the same group, plasma calcidiol levels and statin treatment were inversely associated with the risk of developing the primary target, whereas non-HDL cholesterol levels, treatment with dihydropyridines and verapamil were directly associated (Table 3A and Figure 1A).

On the other hand, variables associated with the development of the primary end point in the group with a NT-proBNP > 125 pg/mL were age, male sex, history of smoking, high blood pressure, cognitive impairment, Caucasian race, previous ventricular dysfunction, previous stroke, previous CABG, previous AF, previous HF, treatment with P2Y12 inhibitors, statins, insulin, oral antidiabetic drugs, anticoagulants, ACE inhibitors, ARBs, betablockers, nitrates, calcium antagonists, dihydropiridines, diuretics, digoxin, proton pump inhibitors and analytical parameters, such as glucose, total cholesterol, eGFR, non-HDL cholesterol, high sensitivity troponin, NT-proBNP, calcidiol, MCP-1, Galectin-3 FGF 23, PTH and klotho (Table 2B).

In the multivariate analysis for the NT-proBNP > 125 pg/mL group, PTH and NT-proBNP levels, age, previous history of CABG and treatment with insulin, angiotensin receptor blockers, nitrates and proton pump inhibitors were positively associated with the development of ischemic events, HF admission or death during the follow-up, whereas treatment with statins was inversely associated (Table 3B and Figure 1B).

ROC curves were also developed for each of the MM biomarkers that demonstrated statistical significance as predictors of adverse events. For PTH and NT-ProBNP in the group of patients with Nt-ProBNP > 125 pg/mL, the AUC in the first case was 0.647 (95% CI 0.591–0.703) and in the second was 0.578 (95% CI 0.520–0.637). In the group of Nt-ProBN *p* < 125 pg/mL, the AUC for calcidiol was 0.386 (95% CI 0.302–0.470) (Appendix A).

## 4. Discussion

One of the most widely used biomarkers in the field of cardiology is NT-proBNP [20,21], which allows the identification of patients with HF, as well as monitoring their response to treatment [9] and predicting the prognosis of patients with cardiovascular disorders [4,20]. In addition, NT-proBNP plasma levels may predict the response to therapy. In this regard, in the CORONA trial, patients with coronary artery disease and left ventricular dysfunction responded to statin therapy only when NT-proBNP and galectin-3 levels were low, suggesting that, in patients with less myocardial damage, the vascular effects of statins still may improve the prognosis [10]. However, it has never been tested if NT-proBNP levels may influence the predictive power of other biomarkers.

Abnormalities of mineral metabolism have implications in the prognosis of patients with coronary artery disease [17,19]. Vitamin D deficiency is associated with an increase in the incidence of cardiovascular adverse events [15,17]. However, vitamin D is part of a system known as mineral metabolism, which encompasses several other components, such as FGF23, PTH and phosphate that may be also related to the incidence of cardiovascular disease. FGF23 is a phosphaturic hormone that helps the kidney to eliminate phosphate and reduce excessive vitamin D levels. High FGF23 plasma levels have been associated with increased mortality, HF and left ventricular hypertrophy [16,17,18]. Similarly, increased PTH plasma levels are related to hypertension, left ventricular hypertroph and increased cardiovascular events [12,22,23]. Furthermore, the soluble form of klotho, the co-receptor of FGF23, has been associated with antiaging and protective cardio-renal effects [24]. There are also other biomarkers of MM that have shown prognostic value in studies carried out by our group. In the past, we analyzed the role of Galectin-3 and MCP-1, and we saw how elevated levels of these biomarkers were associated with an increase in cardiovascular events [4] as well as these biomarkers improving the ability of the LIPID clinical scale to predict the prognosis of patients with stable coronary artery disease [25].

However, to date, there is no evidence as to whether there is a relationship between mineral metabolism and the prognosis of patients with stable ischemic heart disease according to baseline NT-proBNP values.

Thus, in our study, we divided the population according to the presence of NT-ProBNP plasma levels > 125 pg/mL, the cut-off point for the diagnosis of heart failure [9,19], and we assessed which factors of the mineral metabolism system had predictive value in each of the groups.

Our results demonstrate that, in patients with NT-proBNP ≤ 125 pg/mL, calcidiol along with non-HDL cholesterol plasma levels were independently associated with the risk of cardiovascular events. However, in patients with NT-proBNP > 125 pg/mL, the mineral metabolism biomarker with independent predictive value was PTH, along with NT-proBNP plasma levels.

Based on these results, we can infer that the determination of NT-proBNP levels in patients with stable coronary artery disease allows us to differentiate between two types of patients: those in whom the main problem lies in the vascular tree and others in whom the conditioning element is the myocardium. In this regard, low calcidiol plasma levels have been mainly related to atherosclerosis and vascular calcification, and therefore to the risk of acute myocardial infarction and cardiovascular death [11,17,26,27,28]. Our results establish that, the higher the calcidiol levels in patients at no risk of HF, the lower the risk of developing cardiovascular events. Of interest, vitamin D supplementation has not demonstrated a clear reduction in cardiovascular events in previous studies [29,30]. Our data would support the idea that vitamin D supplementation could be especially beneficial in those patients with coronary artery disease who have normal NT-proBNP levels.

On the other hand, our group has recently demonstrated that, in patients with stable coronary artery disease, PTH is associated with the presence of left ventricular hypertrophy [12]. It is also well-known that increased PTH levels favor the development of HF, suggesting that it is mainly a marker of myocardial rather than vascular damage [19,31]. The present study agrees with this hypothesis, suggesting that in patients with coronary artery disease, high NT-proBNP levels identify those whose prognosis depends especially on left ventricular dysfunction and in this subgroup PTH is the component of mineral metabolism that best marks the prognosis.

The main limitation of this work is that, due to the inclusion criteria, this population has a low percentage of left ventricular dysfunction and previous heart failure. Then, our findings may not apply to populations with more extensive left ventricular damage. Another important limitation is that a full clinical characterization of patients at baseline is lacking (i.e., no echocardiographic data, no data on coronary angio and PCI and no data on arrhythmic risk) and this should be recognized. Additionally, limited information on the influence of different medical strategies during follow-up were given.

## 5. Conclusions

In patients with SCAD and NT-proBNP > 125 pg/mL, PTH and NT-proBNP, which are markers related with ventricular damage, are predictors of poor outcome. In the subgroup of patients with NT-proBNP ≤ 125 pg/mL, calcidiol and non-HDL cholesterol, which are more related to vascular damage, are the independent predictors of poor outcome. Then, in patients with SCAD, NT-proBNP may influence the type of biomarker that should be used for risk prediction and help us to focus on myocardial or vascular protection.

## Figures and Tables

**Figure 1 jcm-11-04153-f001:**
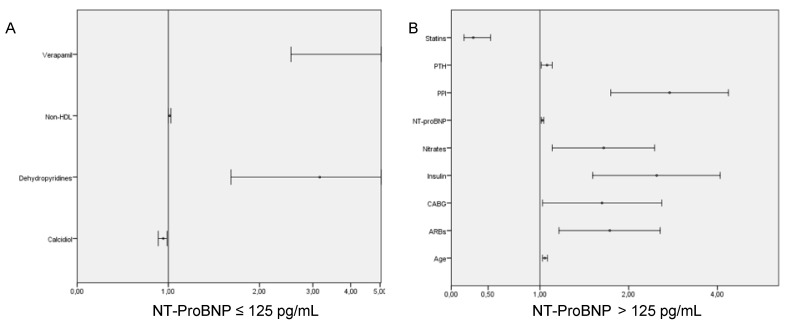
Forest plot graphic showing predictors of death, heart failure admission or ischemic event, according to NT-proBNP levels. (**A**) Group of patients with NT-ProBNP ≤ 125 pg/mL; (**B**) group of patients with NT-ProBNP > 125 pg/mL. ARBs: angiotensin receptor blockers; CABG: coronary artery bypass graft; PPI: proton pump inhibitors; PTH: parathormone.

**Table 1 jcm-11-04153-t001:** Baseline characteristics.

Variable	NT-ProBNP ≤ 125 N = 342	NT-ProBNP > 125 N = 622	*p*
Age (y)	54.0 (48.0–61.0)	65.0 (56.0–75.0)	**<0.001**
Female (%)	17.3	27.3	**<0.001**
Race: Caucasian (%)	95.6	97.4	0.127
Body mass index (kg/m^2^)	28.4 (25.8–30.5)	27.9 (25.6–30.8)	0.359
Smoker (%)	19.3	10.9	**<0.001**
Hypertension (%)	49.1	72.5	**<0.001**
Diabetes (%)	20.8	25.9	0.075
Dyslipidemia (%)	71.9	73.0	0.734
Peripheral artery disease (%)	3.5	3.9	0.784
Cerebrovascular disease (%)	0.3	4.2	**<0.001**
Prior CABG (%)	4.1	10.1	**0.001**
LVEF < 40 (%)	0.9	10.5	**<0.001**
Prior heart failure (%)	2.9	16.4	**<0.001**
Atrial fibrillation (%)	1.5	9.0	**<0.001**
**TREATMENT**			
Aspirin (%)	94.7	92.9	0.273
P2Y12 antagonist (%)	78.1	74.0	0.156
Anticoagulant (%)	1.8	7.6	**<0.001**
Statin (%)	95.9	94.2	0.256
High potency statin (%)	59.4	57.7	0.621
Ezetimibe (%)	5.0	3.7	0.343
Insulin (%)	4.4	7.9	**0.037**
Oral antidiabetic drug (%)	15.5	18.0	0.322
ACEI (%)	59.4	64.3	0.129
ARB (%)	12.3	17.0	0.050
Aldosterone antagonist (%)	2.0	9.3	**<0.001**
Betablocker (%)	73.4	82.2	**0.001**
Diltiazem (%)	3.2	2.4	0.461
Verapamil (%)	0.6	0.2	0.258
Dihydropyridine (%)	12.0	14.8	0.227
Diuretic (%)	12.6	22.2	**<0.001**
Nitrates (%)	10.2	14.5	0.061
PPI (%)	67.5	69.9	0.442
Digoxin (%)	0.0	0.5	0.556
Vitamin D	1.5	1.3	0.779
**PREVIOUS ACUTE CORONARY SYNDROME**			
STEMI/Non-STEMI (%)	39.2/60.8	55.3/44.7	**<0.001**
Number of vessels diseased			
Revascularization method (%)			0.064
• No revascularization	15.8	14.1	
• Drug-eluting stent	56.1	50.2	
• Bare metal stent	23.4	26.4	
• Angioplasty	1.8	3.4	
• CABG	2.9	5.9	
**ANALYTICS**			
Glucose (mg/dL)	100.0 (91.0–114.25)	101.0 (91.7–115.2)	0.401
Total cholesterol (mg/dL)	145.5 (124.0–166.0)	141.0 (124.7–161.0)	0.110
HDL cholesterol (mg/dL)	39.0 (31.4–46.0)	41.0 (35.0–47.0)	**0.029**
LDL cholesterol (mg/dL)	79.0 (65.0–93.0)	76.0 (64.0–92.0)	0.240
Non-HDL cholesterol (mg/dL)	103.0 (85.0–122.2)	99.0 (83.7–116)	**0.048**
Triglyceride (mg/dL)	104.5 (77.5–152.0)	100.0 (76.0–137.0)	0.081
eGFR (mL/min/1.73 m^2^)	87.4 (75.8–97.9)	75.2 (59.2–89.1)	**<0.001**
HsCRP (mg/L)	1.0 (0.3–2.6)	1.1 (0.3–3.2)	0.324
Nt-ProBNP (ng/L)	71.5 (47.2–95.8)	305.0 (187.7–578.0)	**<0.001**
HsTroponin (µg/L)	0.0 (0.0–0.004)	0.005 (0.001–0.014)	**<0.001**
Phosphate (mmol/L)	3.1 (2.7–3.5)	3.2 (2.8–3.5)	0.310
Calcidiol (mmol/L)	19.3 (14.5–25.7)	19.1 (14.2–25.2)	0.779
FGF 23 (RU/mL)	72.7 (56.3–94.9)	82.0 (62.1–108.6)	**<0.001**
Klotho (pg/mL)	588.7 (496.1–730.0)	555.6 (462.7–679.8)	**0.003**
PTH (ng/L)	54.8 (42.1–70.3)	59.8 (45.9–76.9)	**0.001**
MCP-1 (pg/mL)	121.4 (95.2–155.0)	114.7 (114.2–185.6)	**<0.001**
Galectine-3 (ng/mL)	7.2 (5.5–9.0)	8.3 (6.3–108.6)	**<0.001**

ACEi: angiotensin-converting enzyme inhibitors; ARB: angiotensin receptor blockers; BMS: bare metal stent; CABG: coronary artery bypass graft; DES: drug-eluting stent; eGFR: estimates glomerular filtration rate; FGF23: fibroblast growth factor-23; HDL: high density lipoprotein; hsCPR: high sensitivity C-reactive protein; LDL: low density lipoprotein; LVEF: left ventricular ejection fraction; MCP-1: monocyte chemoattractant protein-1; MRA: mineralocorticoid receptor antagonist; Non-STEMI: Non-ST elevation myocardial infarction; NT-ProBNP: N-terminal pro-brain natriuretic peptide; PPI: proton pump inhibitors; PTH: parathormone; STEMI: ST-elevation myocardial infarction.

**Table 2 jcm-11-04153-t002:** Univariate Cox analysis with a significance of *p* < 0.2 according to Nt-ProBNP. (**A**) Patients with Nt-ProBNP > 125 pg/mL; (**B**) patients with Nt-ProBNP > 125 pg/mL.

Variable	HR	*p*
(**A**)		
Caucasian race	0.36	0.093
High blood pressure	1.61	0.123
Statins	0.50	0.184
Ezetimbe	2.55	0.049
MRA	3.25	0.051
Betablockers	0.50	0.025
Nitrates	1.69	0.162
Verapamil	7.47	0.006
Dehydropiridines	2.71	0.002
Cholesterol levels	1.01	0.042
LDL cholesterol	1.01	0.048
Non-HDL	1.01	0.031
Calcidiol	0.95	0.018
PTH	1.01	0.023
(**B**)		
Age	1.06	0.037
Male sex	0.75	0.102
Smoking	0.60	0.143
High blood pressure	2.15	0.001
Cognitive impairment	1.63	0.006
Previous stroke	1.91	0.049
Previous CABG	1.85	0.008
Previous heart failure admission	2.15	0.000
Atrial Fibrillation	1.97	0.004
Left ventricular dysfunction	1.56	0.060
P2Y12 inhibitors	0.75	0.104
Anticoagulation	2.13	0.002
Statins	0.32	0.000
Insulin	2.96	0.000
Oral antidiabetic drugs	28.16	0.002
ACE inhibitors	0.73	0.070
ARB	1.91	0.001
MRA	1.69	0.043
Betablockers	0.61	0.008
Nitrates	2.34	0,000
Diltiazem	2.71	0.006
Verapamil	0.71	0.050
Dehydropiridines	1.36	0.146
Diuretics	1.93	0.000
PPI	2.16	0.000
Digoxin	3.01	0.122
Glucose levels	1.00	0.014
Cholesterol levels	1.00	0.139
Glomerular filtration rate	0.98	0.000
Non-HDL cholesterol	1.00	0.162
High sensitive troponin	1.40	0.054
Nt-ProBNP ^1^	1.02	0.000
Calcidiol	0.99	0.055
MCP-1	1.01	0.004
Galectin-3	1.05	0.001
FGF23	1.00	0.000
PTH ^2^	1.09	0.000
Klotho	0.99	0.073

^1^ Per 100 units of Nt-ProBNP. ^2^ Per 10 units of PTH. Abbreviations as for Table 1.

**Table 3 jcm-11-04153-t003:** Cox multivariate analysis for predictors of adverse events as a function of Nt-Pro-BNP. (**A**) Patients with Nt-ProBNP ≤ 125 pg/mL; (**B**) patients with Nt-ProBNP > 125 pg/mL.

Variable	HR	CI	*p*
(**A**)			
Verapamil	11.28	2.54–50.00	0.001
Dihydropyridines	3.16	1.63–6.13	0.001
Non-HDL cholesterol	1.01	1.00–1.02	0.014
Calcidiol	0.96	0.92–0.99	0.045
(**B**)			
Age	1.04	1.02–1.06	<0.001
CABG	1.62	1.02–2.59	0.041
Statins	0.32	0.19–0.53	<0.001
Insulin	2.49	1.51–4.09	<0.001
ARB	1.73	1.16–2.56	0.007
Nitrates	1.65	1.10–2.45	0.014
Proton pump inhibitors	2.75	1.74–4.36	<0.001
Nt-ProBNP ^1^	1.02	1.01–1.03	<0.001
PTH ^2^	1.06	1.01–1.10	0.011

^1^ Per 100 units of Nt-ProBNP. ^2^ Per 10 units of PTH. Abbreviations as for Table 1.

## Data Availability

The data presented in this study are available on request from the corresponding author. The data are not publicly available due to privacy.

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
