# Peer review of "NT-proBNP Levels Influence the Prognostic Value of Mineral Metabolism Biomarkers in Coronary Artery Disease"

_jcm, 2022, doi:10.3390/jcm11144153_

Round 1

Reviewer 1 Report

In this original paper, Juan Martinez-Milla and collaborators aimed to highlight how the NT-proBNP (using the internationally accepted cut-off of 125 pg/mL) can influence the prognosis value of some mineral metabolism biomarkers in stable coronary artery disease. There is an adequate number of enrolled patients, the design has an interesting foundation, however, there are several issues to be detailed further:

Major comments:

-          In order to outline the relevance of NT-proBNP compared to the MM biomarkers, drawing a ROC curve is an essential step. Furthermore, from those ROC curves, some relevant cut-off values could be extracted (e.g. Youden’s index, which could be then compared to the risk of adverse events).

-          Dividing the patients strictly based on 125 pg/mL cut-off value is not an adequate approach, given the multiple confounding factors, such as age, obesity, renal function, and inflammatory status, to mention just some important aspects that can significantly alter the serum levels of NT-proBNP, with subsequent reporting bias. The statistic must be adjusted accordingly, otherwise the results of the study are not reproducible.

-          The discussion section must be improved with several data from literature, eventually concerning other cardiac biomarkers and their influence/relationship with MM markers and prognosis, respectively. There are many promising biomarkers, such as ST2, GDF-15, galectin-3, which are directly associated with poor prognosis.

-          What would be the effective result of implementing these findings in daily clinical practice? How it can modify the patients’ outcome?

Minor comments:

-          Line 59, a “,” is needed after “Of these…[, ]NT-proBNP…”

-          In several lines, please change the font of NT-proBNP (oversized, e.g. line 153, 156).

-          Check the spelling for the natriuretic peptide: the correct form is “NT-proBNP” (no Nt-proBNP, no NT-ProBNP- multiple forms to be found in text).

Kind regards,

The reviewer

Reviewer 2 Report

The objective of this study was to assess the influence of NT-ProBNP on the prognostic role of MM markers in patients with SCAD. To this end, the authors compared the prognostic value of parathormone [PTH], klotho, phosphate, calcidiol [25-hydroxyvitamin D3], and fibroblast growth factor-23 between patients with NT-ProBNP>125 pg/ml vs patient with NT-ProBNP≤125 pg/ml included in five hospitals of Spain. The main outcome was the combination of death, heart failure, and ischemic events (any acute coronary syndrome, ischemic stroke, or transient ischemic attack). In the group of NT-proBNP>125 pg/ml patients were older, and there were more females and smokers than in patients with normal NT-proBNP. Also, the proportion of patients with hypertension, atrial fibrillation, ejection fraction<40%, cerebrovascular attack, or prior coronary artery by-pass graft was higher in the high NT-proBNP group. In high NT-proBNP pts., the predictors of poor prognosis were PTH and NT-proBNP, along with age, prior coronary artery by-pass graft, treatment with statins, insulin, angiotensin receptor blockers, nitrates, and proton pump inhibitors. In the NT-proBNP≤125 pg/ml subgroup, poor prognosis predictors were plasma levels of non-high-density lipoprotein (non-HDL) cholesterol and calcidiol, as well as treatment with verapamil, and dihydropyridines.

This is a very interesting study that allows one to conclude that in patients with SCAD and NT-ProBNP>125 pg/ml, PTH and NT-ProBNP, that are markers related with ventricular damage, are predictors of poor outcome, whereas in the subgroup of patients with NT-ProBNP≤125 pgm/l, calcidiol and non-HDL cholesterol, more related to vascular damage, are the independent predictors of poor outcome. 

The authors might want to improve the quality of their manuscript widening the section on Limitations. As a matter of fact, a full clinical characterization of patients at baseline is lacking (i.e. no echocardiographic data, no data on coronary angio and PCI, no data on arrhythmic risk, etc) and this should be recognized. Also, poor information on the influence of different medical strategies during follow-up are given.

Round 2

Reviewer 1 Report

The authors honestly addressed my previous concerns, so, from my point of view, it is a significant improvement of the manuscript.

Best regards,

The Reviewer

Author Response

Thank you very much

Reviewer 2 Report

The authors have revised the manuscript accurately.

No further action is required.

Author Response

Thak you very much